# Static and Dynamic Biomaterial Engineering for Cell Modulation

**DOI:** 10.3390/nano12081377

**Published:** 2022-04-17

**Authors:** Hyung-Joon Park, Hyunsik Hong, Ramar Thangam, Min-Gyo Song, Ju-Eun Kim, Eun-Hae Jo, Yun-Jeong Jang, Won-Hyoung Choi, Min-Young Lee, Heemin Kang, Kyu-Back Lee

**Affiliations:** 1Department of Interdisciplinary Biomicrosystem Technology, College of Engineering, Korea University, Seoul 02841, Korea; goodjoon@korea.ac.kr; 2Department of Materials Science and Engineering, College of Engineering, Korea University, Seoul 02841, Korea; legendwing@korea.ac.kr (H.H.); thangam1985@korea.ac.kr (R.T.); 3Institute for High Technology Materials and Devices, Korea University, Seoul 02841, Korea; 4Department of Biomedical Engineering, College of Health Science, Korea University, Seoul 02841, Korea; songmj0919@naver.com (M.-G.S.); jds8203645@naver.com (W.-H.C.); mydevil7@korea.ac.kr (M.-Y.L.); 5Department of Biomedical Engineering, College of Engineering, Korea University, Seoul 02841, Korea; idsinle@korea.ac.kr (J.-E.K.); eunhaejo65@gmail.com (E.-H.J.); 6Department of Biomedical Engineering, Armour College of Engineering, Illinois Institute of Technology, Chicago, IL 60616, USA; yjang12@hawk.iit.edu

**Keywords:** biomaterial engineering, cell modulation, static modulation, dynamic modulation, biomedical engineering

## Abstract

In the biological microenvironment, cells are surrounded by an extracellular matrix (ECM), with which they dynamically interact during various biological processes. Specifically, the physical and chemical properties of the ECM work cooperatively to influence the behavior and fate of cells directly and indirectly, which invokes various physiological responses in the body. Hence, efficient strategies to modulate cellular responses for a specific purpose have become important for various scientific fields such as biology, pharmacy, and medicine. Among many approaches, the utilization of biomaterials has been studied the most because they can be meticulously engineered to mimic cellular modulatory behavior. For such careful engineering, studies on physical modulation (e.g., ECM topography, stiffness, and wettability) and chemical manipulation (e.g., composition and soluble and surface biosignals) have been actively conducted. At present, the scope of research is being shifted from static (considering only the initial environment and the effects of each element) to biomimetic dynamic (including the concepts of time and gradient) modulation in both physical and chemical manipulations. This review provides an overall perspective on how the static and dynamic biomaterials are actively engineered to modulate targeted cellular responses while highlighting the importance and advance from static modulation to biomimetic dynamic modulation for biomedical applications.

## 1. Introduction

Numerous cell responses, such as adhesion, morphology, spreading, arrangement, migration, proliferation, differentiation, and apoptosis, are determined by intricate physical and chemical cellular environments [1]. Specifically, the native extracellular matrix (ECM) is a complex network composed of cell-specific macromolecule multidomain [2]. The components of the ECM are intertwined to create structurally stable composites with physical characteristics such as stiffness and topography that guide cellular responses [3,4]. Cells recognize the mechanical properties of the ECM and convert them into biochemical signals through mechanotransduction to elicit specific cellular responses [5,6,7,8]. Not only the mechanical properties of the ECM but also the bioactive molecules within it, such as growth hormones, cytokines, and ligands, contribute to cellular chemical modulation [9]. To elaborate, the ECM serves as a storage facility for bioactive molecules, thereby regulating numerous soluble factors. Through the regulation of cellular adhesive motifs (e.g., the arginine-glycine-aspartic acid (RGD)-ligand), the ECM also controls cell behavior by serving as a cell-binding site [8,10]. In addition to the ECM, chemical properties, such as pH, concentrations of various metal ions (e.g., Na^+^, K^+^, Ca^2+^, Zn^2+^, Mg^2+^), partial pressures of gases (e.g., O_2_, CO_2_), and gas-related chemicals (e.g., reactive oxygen and nitrogen species), and characteristics of the physical environments, such as temperature, force (e.g., compression and expansion), and energy sources (e.g., light and magnetic and electric fields), also contribute to the regulation of cellular behavior. The outcomes from many studies show that cellular responses can be regulated by artificially creating external environments at the molecular level [11,12,13,14,15]. In other words, since various biochemical reactions within the human body can be induced by the microenvironment formed by artificially applied materials, controlling the physicochemical properties of materials is an important topic in biomedical device and tissue engineering [11,16,17,18,19,20].

Figure 1 shows the changing trends in biomaterials research. Biomaterials are defined as materials used for medical purposes that comprise substances, surfaces, and structures that either do not interact with biological systems or do so cooperatively, and they have been widely used in both diagnosis and treatment [21,22,23]. Most early biomaterials were simple and bioinert, with biocompatibility being regarded as the most important property [24,25]. However, as the secrets of the ECM have been uncovered and the need for more biologically cooperative biomaterials for tissue engineering has increased, bioactive biomaterials with specific functionality have emerged [26,27,28,29,30,31]. In recent years, biomimetic materials have been in the spotlight, as the concept of bio-responsive materials has emerged from using materials to interact dynamically with the human body [32,33,34,35]. The scale has decreased from macro to micro to nano [36], while the complexity of the interactions has increased. Material scientists that used to produce single-function materials are now focusing on multifunctional materials. Moreover, although 1-dimensional (1D), 2D, and 3D materials have already been developed, 4D materials whose characteristics change over time are now emerging [37,38,39,40,41,42].

Biomaterials can be classified as physical-oriented or chemical-oriented, depending on their effects on cells. They can then be further classified as static or dynamic biomaterials. This review provides an overall perspective on the current research into biomaterials to inspire future biomaterials nanoengineering development for controlling the bioactivity of cells (Figure 2). Markedly, it is important to underline that the biomaterials are now advancing from simple static modulation using low dimension (1D or 2D) materials to dynamic modulation with high dimension (3D or 4D) materials that precisely mimic the complex native biological environment. Limitless research and a combination of physical and chemical modulation with dynamicity can significantly advance the field of biomedicine, thereby presenting numerous methodologies and enhancing practical applications in clinics.

## 2. Physical Modulation

### 2.1. Static Modulation of Physical Environment

In vivo, cells are surrounded by the ECM with changeable biophysical properties such as topography and stiffness that can be affected by external forces such as fluid shear stress, compression, and stretching [43,44]. Static modulation refers to the control of cellular responses through the artificial manipulation of various factors, such as topography, stability, and external stimulation in vitro, and the artificial ECM for this is created through various technologies, such as materials engineering, biotechnology, microtechnology, and nanotechnology.

#### 2.1.1. Static Topography

One of the most basic ways to control cellular responses with an artificial ECM is to modify topography [45]. The ECM comprises an interstitial matrix, which is a gel consisting of polysaccharides and fibrous proteins that fills the interstitial spaces in the body, and a basement membrane, which is a sheet-like fibrous network of proteins beneath the epithelium that supports the functions of epithelial cells, including stem cells. The ECM forms different microenvironments depending on the tissue and cell type in the body, and the type of ECM architecture varies from homogeneous meshwork to fibrillar scaffolds. Furthermore, cell adhesion, morphology, production, differentiation, and migration are affected by the local geometry of the ECM [38]. In this section, we focus on the physical topographical features and leading methods for forming micro- and nanotopology and their effects on cells. Changes in topography induce cellular changes through membrane receptors, the cytoskeleton, filopodia, and intracellular signal transduction [14,46]. According to Miyoshi et al. [46], its effect on cell morphology varies depending on the scale of the topographical structures. The topographical scale of the ECM in the tens of micrometers affects the behavior of cells at the singular or multicellular level by changing their peripheral curvature, which, in turn, promotes or restricts the formation of actin fibers depending on the shape. Which ranging from sub-micrometer to 10 μm affects actin cytoskeleton control related to contact guidance, as explained in the next paragraph. Which ranging between 10 and hundreds of nanometers affects integrin clustering. The spaces between the nanostructures control the dissolution and restoration of actin filaments by adjusting the number of molecules of integrin clusters used in the mechanical connections between the ECM and the cells depending on the size.

Technologies for producing varying scales of topography on the ECM in cell culture environments include photolithography, hot embossing, and electrospinning [14,47]. Complicated and complex 3D structures can be produced through additional processing, such as etching, deposition, and imprinting [48]. Structures formed through these techniques have various shapes, such as pillars, ridges, pits, holes, and grooves. Cell adhesion and differentiation can be controlled through these structures, and cell patterning can also be induced by limiting the number of cell adhesion sites. The degree or direction of the cell migration can be adjusted through directional structures, such as grooves, ridges, and lines; the phenomenon through which cells are induced to align and move along the direction of anisotropy is called contact guidance [49].

Meanwhile, cellular response guidance through topographical control has been approached in various ways, including ECM imitation with fibrous structures. For example, Berry et al. [50] used photolithography to produce a regular pit arrangement with increasing spacing and diameter and create a patterned 3D network structure to culture fibroblast cells. Their proliferation varied depending on the diameter of the pit, the angle of the circumference, and the spacing between the pits. Although the surface with the smallest pits (7 μm in diameter) showed the highest proliferation rate, it did not differ significantly from those with pits of other sizes.

Ray et al. [51] produced ridge/groove structures with constant width and spacing using ultraviolet (UV)-assisted capillary force lithography. When carcinoma cells were cultured on the surface, they became more elongated on the aligned pattern than on a flat surface and became highly oriented along the alignment direction. In addition, Kim et al. [52] cultured fibroblasts by making gradient ridge/groove pattern arrays with a ridge width of 1 μm and graded the spacing with UV-assisted capillary force lithography. The fibroblasts became more strongly aligned and elongated parallel to the ridges in proportion to the density of the ridge pattern. However, the migration rate was the fastest in the intermediate part, with a spacing interval of 5.6 to 6.9 μm.

Nomura et al. [53] cultured cells on 1 μm-high nanopillar structures made using photolithography. Compared to a control group grown in a commercial culture dish, cells cultured on the nanopillar structures showed spherical morphology and aggregated to form spheroids, while the cell area decreased, and cell detachment occurred more easily. Bae et al. [54] produced graded-diameter nanopillar arrays using a porous anodic aluminum oxide (AAO) mold by adjusting the pore-widening time. They screened a surface suitable for human embryonic stem cell (hESC) maintenance without a feeder cell layer on the varying topography. Unlike the control group, where the epithelial–mesenchymal transition process occurred due to single-cell spreading, a more compact colony was formed in the smaller nanopillar diameter range, and the expression of undifferentiated markers was high.

Graded-diameter nanopillar arrays have also been inversely applied to form graded pore-array patterns. Kim et al. [55] fabricated gradient pore patterns using two-step imprinting. The graded pore-array patterns on the surface of polystyrene were formed by imprinting with poly(methyl methacrylate) pillar array pattern molds fabricated by imprinting with AAO molds, such as those of Bae et al. [54]. They showed that inducing pancreatic islet-like cluster formations from hESCs was successful on surfaces containing pore arrays with pores of 200–300 nm in diameter but not on ones with pores of 100–200 or 300–400 nm in diameter. In a study on hole structures conducted by Choi et al. [56], holes with varying diameters were produced using master molds with increasing area and depth. Afterward, the size of embryonic stem cells in the holes was restricted, resulting in the formation of an embryoid body. Moreover, stem cells cultured in larger holes caused more cardiogenesis and neurogenesis. Sisson et al. [57] used glyceraldehyde and genipin as crosslinking reagents to fabricate electrospun fibrous gelatin scaffolds that were nontoxic and dissolution-resistant and confirmed that cell proliferation was increased in these ECM-mimicking structures.

#### 2.1.2. Static Stiffness

Stiffness is one of the most basic physical features of the ECM that affects cells [19,58,59]. It depends on the chemically crosslinked or physically bonded fibrous proteins and the density of glycosaminoglycans and is varied by the influence of these biomolecules in each type of tissue in the body [59]. Mechanical changes in the extracellular environment, such as tension and stiffness, are detected by integrin clusters recruiting focal adhesion kinases (FAKs). Afterward, talin, vinculin, paxillin, and adaptor protein p130Cas dock together, thereby inducing mechanotransduction, which transmits mechanical signals from integrin to the actin component of the cellular cytoskeleton [60,61]. Protein expression and translation are affected by this process, resulting in cell cytoskeleton remodeling, cell spreading, and differentiation [61]. Stiffness control is mainly performed by modulating the crosslinker, density, and aspect ratio [62,63].

In a cell-substrate stiffness control study conducted by Deroanne et al. [64], they increased the concentration of bis-acrylamide (a crosslinker) from 0.06% to 0.25%, on which they then cultured endothelial cells. By decreasing the crosslinker concentration, the substrate became soft, and cell adhesion decreased due to decreases in actin and focal adhesion plaque expression. In addition, more endothelial cells were converted to a tube-like pattern.

Another method of adjusting stiffness is substrate topography. Trichet et al. [65] observed that fibroblast cells migrate and change their orientation according to different stiffnesses in a graded polydimethylsiloxane pillar structure with varying spring constant values due to its aspect ratio. Upon the sudden application of a large traction force at the step boundary, cells moved from the soft to the stiff side and became polarized in the direction perpendicular to the boundary.

#### 2.1.3. Static Environment

Topography and stiffness are basic physical elements of the ECM that have been studied for a long time. Meanwhile, cell modulation can be induced in a static environment by applying external stimuli, such as an electric or magnetic field or a specific temperature or pH value, to control cell functions in the body.

Cells are in homeostasis in vivo and interact with the surrounding environment through receptors and ion channels in the cell membrane that deliver chemical, mechanical, and electrical signals that originate from both inside and outside of the cell. When an electric field is applied to a living cell, changes in ion flow through the ion passages via interaction with charged molecules in the cell membrane, changes in the membrane receptor distribution, or direct penetration of the stimulus into the cell to interact with charged entities in the cytoplasm can occur, which can be used to control cellular responses [66,67,68]. Since the most representative example of electric signal use in the body is the neural system, many studies on applying electric fields to neural cells have been conducted. Kobelt et al. [69] found that when neural stem progenitor cells were cultured while a DC voltage was applied for a certain period, neurites grew to a length of around five times longer than those in the control group.

Weak magnetic fields are generated in vivo by current-generating body tissues, such as the heart, brain, and muscles. Studies on measuring biomagnetism to explain body functions or using it as a means of diagnosing diseases are ongoing [70,71]. This magnetic field generated in the body is weak (around a millionth of the Earth’s magnetic field), and thus, it does not significantly affect cells. However, if a magnetic field above the threshold is applied to some extent, it affects cells and the extracellular environment. For example, collagen, a main constituent of the ECM, arranges itself in the direction perpendicular to a static magnetic field due to its negative diamagnetic anisotropy characteristics [72]. When cells and collagen are cultured together under a strong magnetic field, collagen is arranged in the direction perpendicular to the magnetic field, and the cells are aligned in the same direction as the collagen via contact guidance [73]. However, Iwasaka et al. [74] and Kotani et al. [75] found that cells were aligned in the direction parallel to the magnetic field when cells were cultured without collagen. The authors speculated that actin fibers and microtubules (flexible intracellular macromolecules) are aligned according to the torque force applied by the diamagnetic anisotropy. As such, the orientation of the cell can be adjusted by applying a static magnetic field.

Since the human internal environment is maintained at 36.5 to 37.0 °C, cell culturing of mammalian cells in vitro is usually performed at 37 °C [76]. Cellular behaviors change in response to temperature variation. A representative example of this is heat shock protein (HSP), which has chaperone activity that inhibits the aggregation of denatured proteins and helps in refolding to reduce cellular damage caused by heat or denaturing stress [77]. Therefore, it is widely used in cell condition monitoring, immune reaction control, cancer therapy, etc. [78,79,80,81,82]. For example, when cells are cultured at 40–47 °C, they express HSP through the thermotolerance induction process, and HSP protects them from apoptosis and necrosis, thereby enabling their survival. Conversely, Viano et al. [83] cultured keratinocytes at 33 °C and discovered enhanced mitochondrial activity along with significantly decreased cellular proliferation, suggesting that the environmental temperature affects these two processes in human skin.

In general, the pH of the human body is maintained between 7.2 and 7.5, which is optimal for most types of cells, and cells strive to maintain this range [84,85,86]. Changes in pH affect many cell processes, including metabolism, membrane potential, cell growth, material transfer through the cell membrane, the polymerization state of the cytoskeleton, and muscle cell shrinkage [87]. Sharpe et al. [88] cultured keratinocytes and fibroblasts in different pH ranges and found that the optimum pH range for their proliferation was 7.2–8.3. They reported that keratinocyte proliferation and migration occurred at a high pH level, whereas differentiation occurred at a low pH level. Likewise, cellular responses, such as proliferation, migration, and differentiation, change according to the pH of the surrounding environment.

Surface wettability is another important factor influencing cellular behavior in response to biomaterials. This parameter can affect the structures of adsorbed proteins and even cell–substrate interaction. Koc et al. [89] reported that there was a significant correlation between the wettability of the surface of a material and the corresponding protein adsorption and that the most abundant protein adsorption occurred on the surface of microscale superhydrophobic surfaces. Moreover, Lourenço et al. [90] reported that wettability varies depending on the topography of the biomaterial surface and eventually affects protein adsorption on the material surface, which can have a huge impact on cell adhesion and viability. Since various physical factors on the surface of biomaterials also change wettability, they must be carefully considered before designing the surface of a biomaterial.

### 2.2. Dynamic Modulation of Physical Environment

Unlike static modulation, during which the culture environment is maintained unchanged during the cell culture process, dynamic modulation refers to methods in which the concept of time is artificially applied to 2D or 3D culture environments to cause change through external stimulation at any time (Figure 3). In vivo, the ECM’s biophysical and biochemical cues are transmitted by the cell’s signaling system, and cells actively respond accordingly. Therefore, dynamic modulation is a step further toward imitating the actual internal environment in the human body (Figure 3B). Culturing cells in vitro, which simulates the inner environment of the human body, enables greater transparency in research when looking for morphological and functional changes [91], preparing cell cultures for transplantation, and producing disease models for drug screening (Figure 3C). Several factors make the cell culture environment dynamic, and even when the same stimulus is used, the response can vary due to the type of biomaterial, the method used, and the corresponding changes in the cellular environment.

Various macroscale stimuli due to movement of the lungs via breathing, electrical muscle stimulation, the beating of the heart, vibrations from the vocal cords, and the flow of blood occur in vivo. Moreover, small physiological cues on the nanoscale are transmitted through mechanotransduction. Although perfectly reproducing the physiological environment is difficult, links between external stimuli and cell responses can be found by applying multiple stimuli and observing the cellular responses. External stimuli, such as electric fields, mechanical deformation, light, magnetic fields, and temperature, that dynamically modulate the physical environment during cell culturing without damaging the cells can trigger changes in cellular behavior.

#### 2.2.1. Dynamic Topological and Stiffness Changes

Photoresponsive materials can absorb light at various wavelengths (from the visible to the UV regions), which changes their physical properties. Hydrogels are mainly used as photoresponsive cell culture substrates in which crosslinking or degradation occurs in response to exposure to light, and thus, the extracellular environment can become stiffer or softer accordingly. In a study by Yang et al. [93,94], YAP and RUNX2 in human mesenchymal stem cells (hMSCs) were either reversibly or irreversibly activated when a stiff photodegradable hydrogel became soft in response to photo-illumination depending on how long they had previously been cultured on a stiff substrate. Guvendiren et al. [94] observed the short- and long-term effects in hMSCs grown on a hydrogel stiffened by light-mediated crosslinking. After stiffening, the cell area and traction force increased significantly within minutes to hours (short-term), and the direction of differentiation changed after a few days to weeks (long-term). Based on the period of culturing before or after stiffening, hMSCs on a hydrogel stiffened earlier underwent adipogenic differentiation, whereas those on one stiffened later underwent osteogenic differentiation. In addition, unlike the irreversible stiffening/softening of a hydrogel, photoreversible stiffening can subject cells to cyclic mechanical loading by alternating light exposure and darkness. In a study by Liu et al. [95], myofibroblast transdifferentiation was promoted through an increase in smooth muscle α-actin (αSMA) and periostin gene expression in fibroblasts to which cyclic mechanical stimulation was applied via reversible stiffening. Similarly, Rosales et al. [96] repeatedly reversibly photoinitiated crosslinking and photodegradation of a hydrogel, which altered the behavior of hMSCs grown on a soft substrate and a stiff substrate. When the cell culture substrate was changed from stiff to soft, the cell area was decreased, roundness was increased, and the nuclear localization of YAP/TAZ (a mechanosensing marker) was decreased and deactivated. On the other hand, the opposite changes occurred when the substrate was stiffened due to photopolymerization. Kloxin et al. [97] produced declination along either the x- or y-axis of a 2D substrate by applying an irradiation gradient to a photodegradable hydrogel without changing its overall rigidity or along the z-axis of a 3D substrate through flood irradiation. They observed a change in cell spreading behavior through hMSC cultures according to the linear degradation.

A typical example of thermo-responsive dynamic topology control is the use of a shape memory polymer [98]. This is a smart polymer that remembers its initial structure, and when an external stimulus is applied, it returns from a temporarily deformed structure to the pre-determined permanent shape. For example, Davis et al. [97,99] observed that when a shape memory polymer substrate was changed from a temporary micro-grooved topology to the original flat surface due to a temperature change, aligned fibroblast cells became randomly oriented. Raczkowska et al. [100] controlled roughness through multipolymer grafting responsive to temperature and pH.

#### 2.2.2. Dynamic Interaction

Temperature is a stimulus that indirectly modifies the external environment of cells. Poly(N-isopropylacrylamide) (PNIPAAm), a well-known temperature-responsive polymer, undergoes reversible changes with two conformations at around 32 °C (the lower critical solution temperature) [101]. When PNIPAAm is fixed to a solid, its surface polymer chain becomes dehydrated at 32 °C or higher, resulting in higher affinity with the cells, and when the temperature drops, cell detachment is caused by the surface becoming hydrated by water molecules. In this case, the cells only detach from the hydrophilic bottom surface, while the interaction between the cells and the ECM structure is maintained without applying a proteolytic enzyme, a process called cell sheet engineering [101,102]. Yamaki et al. [103] exploited this property of PNIPAAm to immobilize fibronectin on its surface. While the gel was swelling due to the temperature change, the cells did not completely detach, and they stretched equiaxial instead by forming filopodia-like structures in response to mechanical signals rather than the temperature change. Thus, the authors proposed this as a new culturing method for analyzing the mechanical signal transduction of cells. The review of Stetsyshyn et al. shows that protein separation, cell sheet harvesting, and cell separation are possible with steric repulsion caused by multiple external environmental variables, such as temperature and pH, using thermoresponsive and multi-responsive grafted polymer brushes [104,105].

#### 2.2.3. Dynamic Stimulation

Muscles function in response to electrical impulses from the nervous system that induce chemical reactions in the human body, and thus, they grow under original bioelectricity. Electric fields also have a crucial influence on culturing. Matos et al. [106] found that murine neural stem cells proliferate in an AC field of 1 Hz and have improved astrocyte differentiation over neuronal differentiation. In addition, Heo et al. [107] used film electrodes made of graphene and polyethylene to electrically stimulate neural cells in the range of 4.5–450.5 mV/mm for a certain period. When neural cells were weakly stimulated at 4.5 mV/mm, the expression of fibronectin and actin increased and that of vinculin decreased, thereby affecting cell adhesion and increasing cell–cell interaction. Moreover, the cell morphology contracted according to the strength of the electric field. In the case of heart cells, when pulsed electromagnetic field stimulation was applied, the upregulation of cardiac-specific gene expression occurred, and the differentiation of stem cell-derived cardiomyocytes was strengthened [108]. Furthermore, cell alignment and coupling were improved when an electrical signal mimicking heartbeats was applied through a cardiac stimulator to already differentiated cardiomyocytes, which formed an ultrastructural organization seven times larger than that of the control group [109]. In Fonseca et al. [110] study, when multidirectional electric field stimulation was applied to randomly arranged heart cells, the cells were excited by an electric field strength 2–30% lower than when applied in a single direction.

Cell manipulation using magnetic fields, especially in combination with magnetic particles, is being actively attempted. The basic concept is to introduce nontoxic magnetic nanoparticles inside cells to control cellular behavior with magnetic fields. For example, in a study conducted by Du et al. [111], iron oxide nanoparticles internalized by embryonic stem cells were made to form a 3D embryoid body structure by gradually adjusting the magnetic force to remotely apply cyclic mechanical strain. Embryoid bodies made through this process were size-adjustable and had a higher formation success rate than those formed using conventional methods. In addition, gene expression representing differentiation into functional cardiomyocytes was improved by the cyclic mechanical stimulation. Furthermore, the interest in using magnetic force in cell therapy is currently growing. In cell-based therapy, it can be difficult to directly inject therapeutic cells such that they settle down in the damaged area, and even after injection, it is challenging for cells to adhere to the constantly moving target tissue or organ stably. Using a magnetic field not only controls the magnetic particles in the body without damaging tissues but also enables tracking through MRI imaging [112]. In Bos et al.’s study [113], when mesenchymal stem cells labeled with superparamagnetic iron oxide were injected into rats, they could be detected for 7 days in the kidney and 12 days in the liver. Moreover, applying a magnetic field to manipulate internalized magnetic nanoparticles causes movement and weak adherence of cells in suspension [114]. Proof-of-concept studies have been conducted to move magnetized immune cells to the desired location with alternating magnetic fields made using electromagnets [115].

Cells in the human body grow, differentiate, and proliferate in a constantly moving environment. Therefore, it is necessary to imitate the mechanical environment in the human body to study the functions more accurately and/or the reactions of cells in vitro. In 1938, Glücksmann [116] experimented with increasing or decreasing the culturing time after implanting cells into the intercostal muscles in poultry in vitro to study the effects of tension and pressure on osteogenic tissue cultures. Since then, various platforms have been devised to mechanically stimulate cells during culturing. For example, to study cells in organs undergoing constant mechanical motion, such as the lungs, dynamic modulation to apply cyclic mechanical loading to the ECM has been employed in vitro. Huh et al. [117] developed a lung-on-chip device for the drug screening of lung cells cultured with periodically applied pressure-driven stretching to mimic the human alveolar–capillary interface. The findings of this study indicated that cyclic mechanical strain increases the absorption and transport of nanoparticles in the lung. 

In a heart disease model, Shradhanjali et al. [118] cultured cells for transplantation to treat heart disease while dynamically stretching them to mimic the mechanical environment of the heart. In a study by Salameh et al. [119], cardiomyocytes were cultured by applying cyclic uniaxial mechanical stretching to a deformable membrane at 1 Hz. The cells became elongated and oriented in the direction perpendicular to the stretch direction, and the expression of Cx43, a gap junction protein controlling the cell–cell transfer of current, was augmented. Furthermore, Morgan et al. [120] imitated the irregular beating of a healthy heart by applying various frequencies in Gaussian or uniform-random distributed patterns and observed the corresponding cellular responses. Although the viability of cells did not change as the frequency changed, the ratio and form of Cx43 protein expression differed.

In a study conducted by Guo et al. [121], shear stress and compressive force were simultaneously applied using a roller while culturing clinically relevant articular chondrocytes. When only shear stress was applied, cell proliferation increased, and chondrogenic phenotype gene marker expression decreased compared to the statically modulated culture. However, chondrogenic differentiation improved when compressive stress was additionally applied. Cartilage and bone undergo continual compressive stress when a load is applied, and mechanical loading regulates bone remodeling: bone formation increases under mechanical loading, and bone loss occurs when no mechanical loading is applied [122,123]. The effect of mechanical stimulation on bone cells in vitro was reviewed by Ehrlich and Lanyon [124].

Egusa et al. [125] cultured muscle stem cells while applying uniaxial cyclic strain. Unlike in a statically modulated environment, skeletal myogenic differentiation was promoted, and the ECM was transformed when the load was applied perpendicularly. Shear stress, induced by blood flow and circumferential stretching due to varying blood pressure, naturally occurs in blood vessels. Haaften et al. [126] designed a bioreactor that applied these two stimuli independently or integrally and identified a relationship between mechanical stimulation and ECM formation. A clearly preferential orientation of actin fibers was commonly observed when hemodynamic loading was applied. Meanwhile, collagen bundle formation in the ECM under cyclic stretching was suppressed by shear stress application, and collagen markers were found in the cell cytoplasm.

In addition to the above methods, mechanical stress can be applied using longitudinal waves such as ultrasound. Although ultrasound can be applied to 3D structures due to its excellent tissue permeability, its directionality is limited. Orapiriyakul et al. [127] varied the ultrasound amplitude applied to mesenchymal stem cells cultured on a 3D collagen hydrogel scaffold from 30 to 90 nm at 1000 Hz. They observed that when the amplitude was 90 nm, the expression of osteoblast markers such as RUNX2 (runt-related transcription factor 2), osterix, osteonectin, osteopontin, and osteocalcin increased. Tissue-penetrative ultrasound stimuli can be induced through electrical stimulation, and mechanical forces such as compression and vibration generated by ultrasound can affect mechanotransduction. Moreover, the piezoelectric effect (the generation of an electric charge by the application of a mechanical force or vice versa) can be used to induce these effects. Nikukar et al. [128] applied vertical nanoscale stimulation to hMSCs through piezo actuators under a petri dish. Compared to hMSCs under 1-Hz stimulation, those stimulated at 1 kHz showed broader spreading with larger focal adhesions and well-organized cytoskeletal contraction fibers, as well as enhanced expression factors related to osteoblast genesis. 

Examples of physical modulation and cell responses are summarized in Table 1 below.

## 3. Chemical Modulation

### 3.1. Static Modulation of Chemical Environment

Cells are exposed to various chemicals in vivo. In general, most cells live in contact with matrixes or scaffolds, recognize the surrounding environment, and secrete various kinds of molecules into the surrounding space to form their own ECM [3]. The ECM is a 3D network structure consisting of various molecules, including the collagen family, elastic fibers, proteoglycans, adhesive glycoproteins, and glycosaminoglycans. It has a wide range of biochemical characteristics due to the spatial organization, immobilization, and combination of these molecules. It contains proteins such as integrin that can be recognized by cellular receptors, and it is rich in biochemical molecules such as cytokines, chemokines, growth factors, and hormones. These biochemical characteristics of the ECM can control cellular behavior directly through membrane-bound receptor-mediated signaling or indirectly through soluble molecules [143,144]. Biochemical factors can be classified into the surface chemical properties of biomaterials, soluble biosignals, and surface-immobilized biosignals.

#### 3.1.1. Surface Chemical Properties

The surface chemical properties of biomaterials can control both protein and cell adhesion through surface energy or surface charge changes. Proteins interact and fold themselves in response to polar, non-polar, and charged groups on the surface of biomaterials, through which cellular behavior can be adjusted [145,146]. In the early days, the biocompatibility of biomaterials was considered important for their applicability in humans [147], and thus, the focus of the corresponding research was on reducing or increasing simple protein adsorption [148,149]. However, recent research interest has been focused on expanding the chemical properties of biomaterials toward their beneficial interaction with other biomolecules or ligands in the human body [150].

In terms of surface chemical moieties, Wang et al. [151] proved that protein adsorption, cell adhesion, cytotoxicity, blood compatibility, and tissue compatibility vary depending on the chemical formula of the polymer structure. Moreover, Hasan et al. [152] showed that the parameters closely related to cell viability, such as hydrophobicity and protein adsorption, are determined by the types of surface functional groups present on polymer biomaterials. Based on these prior studies, Angelova et al. [153] were able to establish a rationalized polymer selection flow chart for the selection of suitable biomaterials according to the desired application and target.

In terms of surface functional groups, Wang et al. [154] observed various bioreactions, such as attachment rate, presenting morphology, cell proliferation, and neurotrophic functions, of Schwann cells present in nerve tissue according to the type of chemical functional group (methyl, carboxyl, amino, hydroxyl, mercapto, and sulfonic). Their results showed that certain functional groups, such as carboxyl and amino, have a positive effect on the growth of Schwann cells. In particular, the amino functional group induces binding with glycosaminoglycans during cell adhesion because it has a positive charge. Lee et al. [155] proved through comprehensive research that differences in the electrical polarity and wettability of a polymer surface could affect its bioreactivity with cells. However, Lee et al. [156] ascertained that the growth response of Chinese hamster cells to various types of functional groups differs from that of Schwann cells. Thus, it can be inferred that the effects of specific functional groups differ depending on cell type and function.

#### 3.1.2. Soluble Biosignals

Cells are influenced by soluble biomolecules in the ECM, including cytokines, chemokines, growth factors, hormones, nutrients, small molecules (steroids, phenols, salts), reactive oxygen species (ROS), and ions, and vice versa [157,158,159,160,161,162]. Cytokines, growth factors, and hormones directly or indirectly regulate cellular behavior via intracrine, paracrine, autocrine, and endocrine signaling [163,164]. They are either in solution or immobilized via attachment to the proteins or glycosaminoglycans in the ECM and play an important role not only in the functional regulation of cells and the maintenance and repair of tissues but also in pathological processes [165]. 

The addition of soluble biomolecules is achieved by directly adding them to media or attaching them to carriers [166,167], including hydrogels and nanoparticles made of lipids, synthetic biodegradable polymers, natural polymers, and inorganic materials [168,169,170,171]. Kimura et al. used hydrogel carriers and gelatin microspheres to control basic fibroblast growth factor (bFGF) release [172]; bFGF in the microspheres was released via degradation of the hydrogel in water rather than by diffusion, which affected the formation of new adipose tissue. Heparin-functionalized chitosan/poly(γ-glutamic acid) (HP-CS/g-PGA) nanoparticles produced via the self-assembly of poly(γ-glutamic acid), heparin polyanions, and the polycation of chitosan have been used to carry a heparin/bFGF complex [173]. The pH-sensitive HP-CS/g-PGA nanoparticles rapidly decomposed in the repaired tissue at pH = 7.4, and bFGF was released in the ischemic tissue at pH < 6.7, which improved angiogenic tube formation via the proliferation of human foreskin fibroblast cells and umbilical vein endothelial cells. Zandi et al. [174] produced polyelectrolyte complex nanoparticles, which are similar to proteoglycan, as a growth factor nanocarrier containing vascular endothelial growth factor (VEGF). They observed that the metabolic activity of human umbilical vein endothelial cells increased more with the nanocarrier containing VEGF than that without VEGF. 

Releasing methods using soluble factors are simple but have the problems of initial burst release kinetics and rapid clearance. Because biomaterials containing soluble biomolecules mainly release them through diffusion or degradation, poor elution can result in insufficient biomolecules actually reaching the target, while excessive elution can have a detrimental effect on other cells and tissues as well as the target cells. Various methods have been proposed to overcome this problem [168,169,170]. One of the most notable is to overcome the initial burst and achieve zero-order release of soluble substances using an osmotic pump, which is available in capsule form and can be conveniently applied using various soluble factors, not only in vitro but also in vivo. The first modern osmotic pump was reported in 1955 [175], and today, various delivery systems using similar principles are widely applied for the zero-order release of substances for both research and clinical treatment purposes.

#### 3.1.3. Surface-Immobilized Biosignals

Another method to control cell behavior involves using surface-immobilized biosignals on the ECM. In fact, cytokines and growth factors, which are widely used in the field of biomedical engineering, are present not only in body fluids but also at specific binding sites in the ECM. To mimic this process in vivo, numerous studies have been conducted to control cellular behavior by coating or attaching various biomolecules to the surfaces of biomaterials. Surface-immobilized biosignals have the advantages of targeting accuracy and elaborate dosage control, thus overcoming the limitations of using soluble biomolecules. The presentation of chemical biosignals on the surface of biomaterials can be classified into physical and covalent immobilization and ligand-presenting techniques.

Physical immobilization methods can be easily conducted under mild conditions. For example, Koh et al. [176] immobilized biosignals through a physical adsorption method of immersion in a laminin solution of plasma-treated poly(ℒ-lactic acid) nanofibers. Unfortunately, this method produced unpredictable release profiles not significantly different from those produced using soluble biosignals except that they were coated on the material surface rather than the material itself and had the disadvantage of limited biomolecule content. To solve this, layer-by-layer approaches enable spatial and temporal delivery control using bilayered or multilayered coatings [177]. For example, Cruzier et al. [178] used crosslinked poly(L-lysine)/hyaluronan (HA) layer-by-layer films as storage for delivering recombinant human bone morphogenic protein 2 (rhBMP-2) to myoblasts. The rhBMP-2 in the thin film used as an adjustable storage mechanism remained bioactive for 10 days or more. The layer-by-layer approach can also be applied for the delivery of multiple growth factors, as demonstrated by Oliveira et al. [179]. They produced a multilayer of platelet lysate as the source of multiple growth factors using charged polysaccharides as the polyelectrolyte with different sulfation degrees and charges for the formation of multiple bilayers of polysaccharides and platelet lysate using the layer-by-layer approach. This multilayer was evaluated on human adipose-derived stem cell cultures and promoted morphological changes, serum-free adhesion, and cell proliferation.

Covalent immobilization techniques based on surface modification have been attempted via coupling chemistry (e.g., carbodiimide coupling [180], maleimide–thiol coupling [181]), surface brush formation (e.g., surface-initiated atom transfer radical polymerization (SI-ATRP) [182], and photopolymerization [183]. This approach prevents the initial burst release of biomolecules and improves binding stability [184], and it is especially useful when using immobilized biosignals. However, chemical or enzymatic cleavage, together with the release of covalently immobilized biosignals, must be considered. It has the advantages of being able to precisely control the number, orientation, retention, and distribution of the biomolecules emitted, as well as enabling local and continuous delivery. For example, Carbonneau et al. [185] used the carboxylic acid groups in chondroitin sulfate and the primary amine groups in epidermal growth factor (EGF) for the successive carbodiimide coupling of biomaterial-chondroitin sulfate-EGF. The surface increased the adhesion, growth, and resistance to apoptosis of vascular smooth muscle cells in a serum-free medium. Ravi et al. [186] attached RGD-ligand peptides to the surface of an elastin-like protein hydrogel using maleimide–thiol coupling, which improved the attachment, migration, spreading, and provision of endothelial cells. Xu et al. [187] formed poly-((meth)acrylic acid) brushes on the surface of polylactide film via SI-ATRP and attached gelatin via carbodiimide coupling, which improved cell adhesion. Itoga et al. [188] cultured endothelial cells on the surface of a poly(ethylene glycol) micropatterned cover glass using photopolymerization, which achieved cell patterning.

Ligands, the etymology of which includes “ligare” (Latin for binding), form complexes with biomolecules to fulfill myriad biological purposes. Substrates, coenzymes, hormones, and drugs bind specifically to proteins such as enzymes and receptors via ligands. They not only facilitate the binding of molecules with an appropriate inverse structure but also interact with specific receptors present on the surface of cells, tissues, and organs to generate biological signals. Extracellular ligands such as the RGD peptide, fibronectin, fibrinogen, and collagen are recognized by integrins, thereby invoking changes in the cytoskeleton and gene expression that eventually determine cell polarity as well as cellular behavior and fate. Numerous studies show that cellular behavior can be controlled through focal adhesion as well as other downstream signals through ligands on the surface of biomaterials [189,190,191,192]. Based on technologies such as self-assembled monolayers (SAMs) [193] and colloidal lithography, precise and complex surface modification is possible by adjusting not only the type of ligand but also its spacing, ordering, frequency, aspect ratio, and stability.

For example, adjusting the distance of ligands by patterning and fixing the nanoparticles coated with RGD-ligand on a substrate through block copolymer nanolithography is an often-used approach. Changes in cellular responses have been observed after adjusting the distance between the pattern space and ligand sites [194,195,196,197]. On the surface of substrates with controlled RGD-ligand spacing on the tens of nanometer scale, cell adhesion is limited when the interval between the ligand sites is large, whereas stem cell osteogenic differentiation occurs when the interval is small [191,192,193,198]. On the other hand, Wang et al. [194] reported that regardless of the cell spreading size, RGD nano-spacing acts as a strong regulator of cell tension and stem cell differentiation, although they were unable to conclude that this result was due to the micro- or nanopatterns used. Furthermore, Arnold et al. [199] cultured fibroblast cells on substrates by controlling the pattern space while keeping the distance between the ligand sites constant and found that the size of the paxillin domain expanded as the pattern spacing was increased.

In the case of controlling ligand ordering, Huang et al. [200] adjusted the spacing of nanoparticles coated with RGD-ligand on a substrate fixed in either a regular or irregular nanopattern and then cultured cells on it. When the distance between the RGD-ligand sites was less than 70 nm, the influence of the regular pattern was not important. However, when it exceeded 70 nm, the regularity between the ligand sites affected cell adhesion.

Min et al. [201] synthesized ligand-loaded nanorods with tunable ligand recurrence while keeping the overall RGD-ligand density constant and located the ligand sites in or at the edge of the nanorods. The effect of varying the ligand recurrence and location on cellular behavior was studied via culturing stem cells and macrophages on the nanorods, which promoted the focal adhesion-assisted mechanotransduction and differentiation of stem cells with low ligand recurrence. It is worth noting that ligand sites located at the edge of the nanorods also promoted this cellular behavior.

It is also possible to adjust the aspect ratio of the ligand [202] by coating RGD-ligand on Au nanorods exhibiting various aspect ratios. When the macrophages were cultured on the RGD coated Au nanorods, macrophages showed increased adhesion and regenerative (M2) polarization on Au nanorods with a high aspect ratio.

As ligand stability (ligand–substrate binding strength) increases, cell adherence, spread area, and differentiation increase. Choi et al. [203] controlled the concentration of silane when silanizing substrates, which regulated the stability of the electrostatic interactions between citrate-capped Au nanoparticles and the substrate. Stem cells cultured on a substrate with high coupling strength showed increased spreading-like adhesion and osteogenic differentiation in the presence of an induction factor.

### 3.2. Dynamic Modulation of Chemical Environment

Cells and the ECM interact dynamically in real time. However, reflecting this is difficult in vitro when only the initial characteristics of the cell culture environment are considered. Meanwhile, although the exact number of biosignals released and their release time are important, it is challenging to mimic this in vitro. To overcome these limitations, many groups have strived to monitor, identify, predict, and control the cell-mediated dynamic remodeling of the ECM induced by the secretion, degradation, or adsorption of proteins [204]. Similar to the dynamic modulation of the physical environment mentioned above, the chemical environment can also be dynamically modulated through light, electricity, magnetic, and ultrasonic stimulation, as well as self-assembly. The dynamic modulation of the chemical environment can be controlled by the release of soluble biosignals, interactions between cells and biomaterials, or the application of stimuli.

#### 3.2.1. Dynamic Release

As discussed above, in the case of static modulation through soluble biosignals, it is difficult to adjust the release amount and time. Thus, the concept of “dynamic release”, which solves these problems through chemical reactions caused by external stimuli, was introduced.

In the case of dynamic release using light stimulation, a nanocarrier using photoresponsive molecules that cause chemical changes via UV and visible light can be used. For example, in a recent study [205], a nanocarrier for calcium regulation was inserted into macrophages. Near-infrared light was converted into UV light by upconverting nanoparticles, and a photocleavable linker was cut using a photoswitch, which released calcium inside the carrier. When the calcium concentration in the macrophages increased, the macrophages favored polarization toward the M1 phenotype. A nanocarrier using photoresponsive molecules is limited to noninvasive application to epithelial tissues or tissues just below them by light wavelengths easily absorbed by biological tissues.

Dynamic drug release via external electrical stimulation can be achieved by utilizing electrically responsive substrates to release drugs that can alter cellular behavior. For example, George et al. [206] developed a drug delivery platform using a polypyrrole (PPy)-conductive polymer substrate. In this system, drug release was triggered when a potential difference using biotin-streptavidin coupling was applied to a nerve growth factor (NGF)-loaded PPy scaffold, resulting in the neurite outgrowth of PC-12 cells. They also confirmed that NGF was released in response to electrical stimulation pulses of varying lengths. Meanwhile, Luo et al. [207] efficiently loaded dexamethasone molecules into multi-walled carbon nanotubes (MWCNTs) that were then sealed using PPy, after which drug release was initiated using electrical stimulation. This inhibited the activation of highly aggressively proliferating immortalized (HAPI) cells (a microglia cell line) cultured with lipopolysaccharide and decreased the nitrous oxide concentration. In addition, Cui et al. [208] loaded phBMP-4 (plasmid form of bone morphogenic protein-4) into an electroactive poly(lactic-co-glycolic acid)/hydroxyapatite/poly(l-lactic acid)-block-aniline pentamer-block-poly(l-lactic acid) (PLGA/HA/PLA-AP) tissue-engineering scaffold, which was then released through electrical stimulation at a 50% duty cycle. This enhanced the proliferation of osteoblasts as well as osteogenesis differentiation in vitro and resulted in effective bone healing in vivo.

In the case of dynamic release using magnetic stimulation, a ferrogel can be used in a magnetic field. Zhao et al. [209] controlled drug release and cell release from a microporous ferrogel using magnetic field stimulation. The stiffness of the microporous ferrogel was adjusted by varying the concentration of crosslinker, and the size-controllable micropores were produced by mixing gels frozen at different temperatures, followed by lyophilization and subsequent rehydration. To evaluate the drug release efficiency, a drug was added to the resulting microporous ferrogel, and 120 cycles of (on/off) stimulation were applied for 30 min in a magnetic field. As a result, the ferrogel stimulated by the magnetic field released a larger amount of the drug than that without magnetic stimulation. In addition, dermal fibroblast cell release from an alginate ferrogel with an attached RGD peptide showed that the cell release efficiency increases as the RGD density decreases.

Interestingly, some studies mimicked native metabolism for the dynamic release of ions that enabled cellular regulation [210,211]. When such materials were utilized in vivo, they gradually degraded via chemical and physical reactions that occur to facilitate stem cell differentiation in vivo [212].

#### 3.2.2. Dynamic Interaction

Dynamic interaction is a method of controlling cellular behavior, including cell adhesion to substrates, by changing the chemical properties of the surface of the biomaterial. Dynamic interaction modulation through light enables localization and high-resolution control and has the advantage of only slightly affecting untargeted cells. RGD peptides can be protected using UV-based photolabile materials to temporally control cell adhesion [213,214]. UV exposure of RGD sites protected through UV-based photolabile materials promotes cell adhesion and cell migration more than that of unprotected RGD sites [213]. Peptides can be fixed on a photoswitchable material that can be light-triggered by UV [215,216,217]. The structure is deformed by photoinduced isomerization when the photoswitchable material is exposed to UV, and the deformed structure is restored to its original state when exposed to visible light. Through this, a dynamic adhesion environment can be created by adjusting the time point at which the peptide sites on the cell attachment surface are exposed, which affects cell differentiation.

Directly applying an electric force can change the interaction between the cells and the substrate that directs cellular behavior. Zhang et al. [218] applied DC electrical stimulation to human adipose-derived mesenchymal stem cells cultured on PPY/PCL (an electrically conductive scaffold), resulting in a 100% increase in calcium deposition on the substrate. Accordingly, the cells migrated to the inner region of the scaffold, and their osteogenic differentiation was enhanced, thereby posing the possibility that PPY/PCL can be used as a scaffold material for bone healing. Browe et al. [219] created a hydrogel that acts as an electroactive polymer actuator by crosslinking poly(ethylene glycol) diacrylate and acrylic acid and optimized its properties for muscle tissue development. Applying a DC voltage to the hydrogel caused angular displacement as the actuation response, thereby enabling the manipulation of diverse cellular responses, such as metabolic activity, intracellular matrix production, and cell attachment of C2C12 mouse myoblasts, depending on the optimized hydrogel properties.

In the case of magnetic stimulation, it is commonly used in conjunction with magnetic particles with chemical functionality as a magnetic field has the advantages of being highly cytocompatible and tissue-penetrative. In a recent study, Au nanoparticles coated with RGDs were grafted to a substrate, and then a larger-sized magnetic nanoparticle was grafted to the RGD-coated Au nanoparticles with a PEG linker molecule to conceal the underlying ligand sites, thereby enabling their temporal regulation by applying a magnetic force. When the ligand sites were revealed, stem cells cultured on the substrates exhibited increased focal adhesion, spreading, differentiation, and mechanosensing [19]. In addition, studies on nano-switching using metal-ion-ligand complexes have also been conducted [220]. In another study, magnetic nanoparticles were coated with RGD-ligand and grafted to a substrate via the PEG linker molecule, which enabled self-control of the ligand vibration motion [211]. When macrophages were cultured on this substrate and high-frequency vibrations were applied, macrophage adhesion was inhibited, and M1 polarization was promoted, while low-frequency vibrations promoted macrophage adhesion and M2 polarization. Khatua et al. used the magnetic field to dynamically control the density of negatively charged ligand sites on magnetic nanoparticles attached to a positively charged substrate on which stem cells were cultured [221]. On the magnetically attracted high-ligand-density side, stem cell adhesion, mechanosensing, and differentiation were promoted. In a slightly different manner, a magnetic field was also employed to control the RGD-accessibility of the cells via screening and unscreening the buried ligands [222]. When the ligands located on the substrate surface were completely screened via magnetic nanoparticle clusters, macrophage adhesion was hindered, while unscreening of the ligands promoted macrophage adhesion as well as M2 polarization.

Bioactive moieties are used in self-assembly mechanisms. Proadhesive cations such as, ${\mathrm{Mg}}^{2+}$, ${\mathrm{Mn}}^{2+}$, or ${\mathrm{Ca}}^{2+}$ are activated by binding with integrins, which are switchable in situ via their combinatorial assembly, thereby easily controlling cell adhesion and functions. Since ${\mathrm{Mg}}^{2+}$ induces the binding of integrin and cell adhesion [223], cell adhesion and release can be induced according to the presence of a cell-adhesive ${\mathrm{Mg}}^{2+}$ moiety. EDTA chelation with ${\mathrm{Mg}}^{2+}$ bound to Au nanoparticles coated with bisphosphate (BP) on a substrate promotes cell release, while the addition of RGD to the nanoassembly promotes focal adhesion. This mechanism provides a dynamic environment in which cell adhesion and release are temporally controllable. Culturing stem cells on the substrate promotes RUNX2 and ALP expression, signifying stem cell differentiation when RGD nano-assembled with ${\mathrm{Mg}}^{2+}$-BP-Au nanoparticles.

#### 3.2.3. Dynamic Stimulation

In addition to indirect cell modulation through changes in soluble biosignals or surface chemical properties, direct cell modulation through chemistry and stimulation is classified as dynamic stimulation. Both direct and indirect electrical stimulation can be presented to cells via an external electric field to manipulate cellular behavior.

Hanna et al. [224] cultured hMSCs that exhibited spontaneous calcium oscillation upon application of microsecond electric pulses. This was due to Ca^2+^ penetrating the cells, which resulted in either the occurrence or lack of spontaneous calcium oscillation depending on the electric field amplitude. In a study conducted by Sauer et al. [225], cardiomyocytes in an embryoid body were subjected to a single DC field pulse (500 V/m), which resulted in increased intracellular ROS. Since the degree of beating foci differentiation and size both increased with increasing ROS, the authors concluded that ROS affects cardiac development. Wan et al. [226] showed that fibronectin conformation could be independently controlled on the macroscopic scale through electrical stimulation. They cultured 3T3-L1 mouse fibroblasts on a conducting polymer device subjected to varying electric potential levels and then analyzed fibroblast adhesion. In addition, this device enabled the meticulous modulation of protein conformation by altering the electric field, suggesting that the developed model could be used to further understand cell–substrate interaction.

In the case of magnetic stimulation, it is possible to directly stimulate cells using chemically modified magnetic particles [227]. In a study by Yun et al. [228], magnetic nanoparticles were injected into neural stem cells and migrated to brain tissue using a magnet. They identified that cells containing magnetic nanoparticles tended to differentiate more readily into neurons or astrocytes than cells not containing them. Moreover, signal transfer can be activated using magnetic nanoparticles. Mannix et al. [229] and Lee et al. [230] produced magnetic nanoparticles coated with a ligand. The ligand and transmembrane receptors combined with the nanoparticles and became aggregated due to attraction between the nanoparticles in a magnetic field, which resulted in clustering that affected cell signal transduction. Surprisingly, magnetic nanoparticles decorated with ligand arrays under the theoretical inter-distance limit (caused by the inherent repulsion of AuNPs) were reported via unprecedented in situ seed-mediated growth to stimulate stem cell adhesion and differentiation [231]. Zhang et al. [232] produced an array with 1D materials via the self-assembly of magnetic nanoparticles and attached DNA to the array through electrostatic interactions after hydrophilic polymer encapsulation. The DNA-coated magnetic array was introduced into mesenchymal stem cells and then injected into rats. They ascertained that genetic engineering, which was effective for the mesenchymal stem cells in the array, is possible, as indicated by the overexpression of a neurotrophic factor. Du et al. [111] introduced magnetic nanoparticles into embryonic stem cells to provide 3D geometry magnetic stimulation using magnetic microtips. Although they found little difference in gene expression compared to the standard hanging drop method, differentiation of the ESCs to form a mesodermal cardiac pathway was observed. Intriguingly, a magnetic field could be used to modulate the inherent features of a nanohelix [233]. In the study, the inter-distance between the ligand pitch was controlled via magnetically modulated winding and unwinding of the ligand-coated nanohelix, which regulated macrophage polarization.

Direct stimulation via ultrasound is used in cancer therapy. It works with a chemical compound known as a sonosensitizer to create ROS that can kill cancer cells. Giuntini et al. [234] confirmed ROS generation by treating water-containing metal-porphyrin complexes with ultrasound. Li et al. [235] encapsulated hydrogen peroxide ($H_{2}O_{2}$) in iron(II,III) oxide (${\mathrm{Fe}}_{3}O_{4})–\mathrm{PLGA}$ polymersomes that were easily destroyed when exposed to ultrasound. This generated OH through the Fenton reaction between $H_{2}O_{2}$ and ${\mathrm{Fe}}_{3}O_{4}$, which suggests that this could be an effective cancer cell therapy.

It is also possible to directly stimulate cells through self-assembly. In the case of cellular enzyme-regulated self-assembly, Tanaka et al. [236] produced a gelator precursor that could cut through matrix metalloproteinase-7 before entering a cancer cell and then self-assemble within the cell. The hydrogelation stressed the cancer cells, thereby inducing apoptosis. Moreover, Li et al. [237] observed the inhibition of cancer cell proliferation and apoptosis using precursor enzyme-instructed self-assembly via the catalytic reaction of carboxylesterases. Although the material could be remotely controlled without physical stimulation through self-assembly, the self-assembly was irreversible and spatiotemporal control was difficult. 

Examples of chemical modulation and cell responses are summarized in Table 2 below.

## 4. Conclusions and Perspectives

The cellular environment consists of cells and the ECM. Interactions among cells are beyond the scope of this review. Focusing on the ECM, its physical and chemical properties determine cell fate through a variety of mechanisms, such as direct interaction, intracellular signaling, direct nuclear signaling, and mechano-sensitivity signaling (Figure 3).

For example, stem cells exist in a local microenvironment called a niche, and the niche’s biophysical and biochemical properties have a decisive influence on cellular behavior and cell fate. Morphogenesis, organogenesis, self-renewal, differentiation, and maintenance of potential are also determined by the stem cell microenvironment [262]. These biological responses can be explained through mechanotransduction. Cell adhesion phosphorylates FAKs of focal adhesion complexes, which activates mechano-sensitivity signaling. Representative examples include MAPK and transforming protein RHOA. RHOA phosphorylates ROH-associated protein kinase 1 (ROCK), which then phosphorylates myosin light chain (MLC), resulting in non-muscle myosin II activation. The contraction of YAP/TAZ, megakaryocytic acute leukemia (MAL, also known as MRTF-A, MKL1, a G-actin-binding co-activator of serum response factor (SRF)), as well as WNT effector β-catenin results in a cell response [263]. In addition, the mechanical cues of the niche are directly connected to the nucleus through nuclear lamina proteins such as lamin A (LMNA), affecting chromatin structures and causing epigenetic regulation [262].

In particular, epigenetic regulation, which regulates gene expression through DNA methylation and histone modification without changing the basic sequence of DNA, is closely related to the differentiation of stem cells [264]. DNA is surrounded by histone octamers to form a chromatin structure and is bound to the structural and regulatory proteins, and DNA and histone modifications can improve chromatin accessibility in the promoter region. Lamina-associated domains (LADs) interacting with nuclear lamina can be decomposed into regions with high transcriptional activity by stimulation, which closely affects gene transcription. Meanwhile, nucleosomes are composed of four histone proteins (H2A, H2B, H3, and H4) and are gathered in two groups to surround DNA as octamers, and these histone proteins undergo post-translation processes, such as methylation, phosphorylation, acetylation, ubiquitination, and sumoylation. These modifications change the histone structure and affect the possibility of combining transcription factors and regulatory factors associated with co-repressor complexes to regulate transcriptional activation. Through this process, external stimuli can cause chromatin changes that last for a specific time, which is called “cell memory”, one of the concepts that must be considered in the selection of biomaterials [265]. 

In addition to the aforementioned stimuli from the ECM, neighboring cells may also cause specific cellular behaviors through the deformation of the cytoskeleton and the change of ligand–receptor interactions and ion channels through cell-to-cell forces induced by cadherin-catenin complexes. This modification changes the histone structure and affects the possibility of combining transcription factors and co-repressors to regulate transcriptional activation.

Various biomaterials have been developed for the modulation of cellular behavior. The rapid development of new principles in various fields, such as materials science and engineering, biology, pharmacy, and medicine, has enabled the emergence of new biomaterials. In addition, new modalities for the control of cellular behavior, either statically or dynamically, are also being developed [266]. Accordingly, it is necessary to establish the concept of dynamic modulation of the cellular microenvironment. Thus, the “dynamic modulation of the cellular microenvironment” can be defined as “changes in the cellular microenvironment induced by intentionally locating materials, substances, and/or energy fields.”

“Biological materials” refer to all substances made by biological systems that can regenerate and often contain genetic information, whereas “biomaterials” are synthetic or natural materials used for bone and tissue healing or manufacture artificial organs and prostheses [212,267,268,269,270]. Although biomaterials and biological materials are different by definition, the boundary between them is becoming ambiguous due to the development of tissue engineering [240] and regenerative medicine. Indeed, biological materials are increasingly being used as biomaterials [271,272]. Bio-scaffolds, including decellularized ECM scaffolds, functional tissues, and organ-like structures such as cell sheets, are widely used as biomaterials. In addition, in the 3D bioprinting process for manufacturing artificial organs, cell spheroids or organoids are used as biomaterials for building blocks. From an engineering perspective, tissues used for living tissue replacement and tissue grafts, and even donor organs for organ transplantation, can be classified as biomaterials [273].

Biohybrid materials are also worth noting [274,275]. These are compounds made up of both biological substances (e.g., biomolecules, cells, and tissues) as functional units and non-biological substances (e.g., polymers, ceramics, and metals) as structural units [276]. Due to the advantages of biohybrid materials that allow diverse functionality as well as high stability, they are continually being studied and are applicable in the fields of biosensors, biocatalysts, remediation, and therapeutics [276,277,278]. In the future, biomaterials will be not only biocompatible and anti-bacterial but also capable of interacting with the ECM via complex 3D biomimetic mechanisms.

Although studies have been conducted on both the physical and chemical characteristics of biomaterials, neither should be considered independently [159]. Chemical factors can change due to changes in the physical environment (e.g., heat treatment, stoichiometry, polymer chemical composition, and tacticity) and vice versa (e.g., functional modification and chemical component modification). Moreover, cell–ECM interaction and the adsorption of biomolecules such as proteins can also alter the physical and chemical environments [279,280]. For these reasons, biological interactions and changes over time, as well as the combinatory physical and chemical effects of biomaterials, must be considered when constructing an optimal cellular environment and efficiently controlling cell responses.

The importance of multi-stimuli systems and dynamic environmental modulation while considering the complexity of cellular behavior modulation over time will increase with the advancement of biomaterials. However, without achieving an understanding of cell regulation by single factors, discontinuous environments, and static regulation, cell regulation through multi-factors, continuous environment, and dynamic regulation is meaningless due to the complexity of cell behavior. However, studies on the nature of single surfaces, static modulation, and discontinuous environments should still be conducted to achieve a fundamental understanding of cellular behavior modulation. Dynamic modulation through additional stimuli based on static modulation will allow biomaterials to be more biomimetic and bio-responsive.

In the field of tissue engineering, overall modulation through the control of the chemical, static, and dynamic environments provide a basic structural scaffold, which will contribute to the production of artificial skin and cartilage [281] as well as artificial organizations with practical functionalities [282,283,284]. In organ regeneration, it will become the foundation technology that enables the regeneration of organs such as the heart, liver, kidney, lungs, larynx, trachea, and bronchi [285,286,287,288]. In the regenerative medicine field, it is expected to be widely used in not only neurology related to CNS regeneration but also urology [289,290,291,292,293]. In the field of drug delivery, it will not only provide basic delivery devices but also lead to advances in technology related to transdermal and intracellular delivery [170,294,295,296,297,298,299,300,301]. In wound healing, it will provide functional hydrogels and help develop skin graft technology [302,303,304,305]. Moreover, in the dental field, it will contribute to the provision of crowns, bridges, and implants for dental treatment and periodontal regeneration [306,307,308]. Finally, in the field of disease diagnostics, new analysis technologies and innovative devices can be applied to cancer diagnostics, Alzheimer’s diagnostics, noise diagnostics, etc. [309,310,311,312].

Advancements in biomaterial engineering through physical, chemical, static, and dynamic modulation provide fundamental understandings and insights into the complex cellular regulatory mechanisms that will lead to advancement in the fields of biology, nanomaterial science [313], materials chemistry [314], and biomedicine. Followingly, biomaterials are evolving from simple low-dimensional (1D and 2D) static modulation to biomimetic high-dimensional (3D and 4D) dynamic modulation that precisely imitate the intricate physical and chemical native cellular regulatory mechanisms for targeted cell modulation. Furthermore, this will contribute to practical application in clinics, such as new regenerative therapies for tissue and organ replacement, and thus advance the field of biomedicine [315,316]. Still, with numerous regulatory parameters and mechanisms to be divulged, physical and chemical dynamic modulation must be studied intensively for limitless and efficient practical application in clinics.

## Figures and Tables

**Figure 1 nanomaterials-12-01377-f001:**
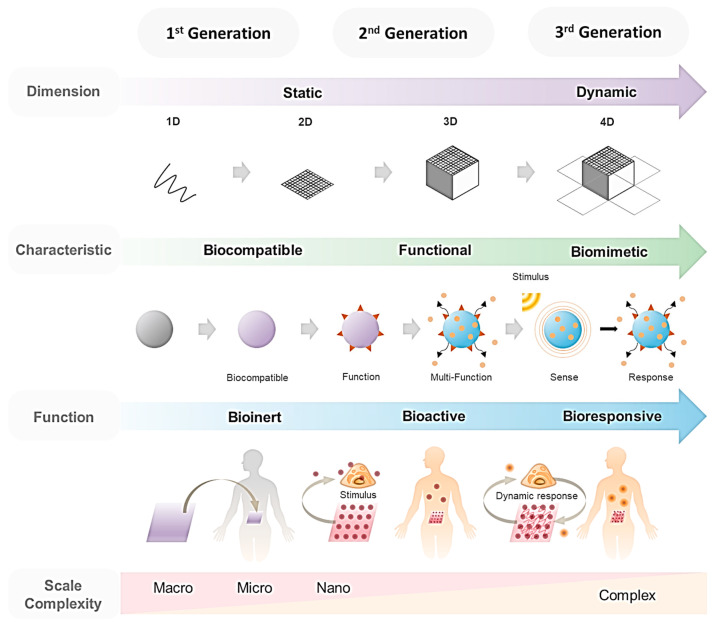
Trends in biomaterials research. Biomaterials can be largely divided into three generations. First-generation biomaterials are bioinert materials, and the focus is on the biocompatibility of the materials themselves. Second-generation biomaterials are bioactive materials that, in addition to being harmless to the body, have specific functions through the physical and chemical modification of the material surface and drug release. Third-generation biomaterials are bio-responsive materials that can organically react with living organisms to surroundings or specific stimuli. In particular, the concept of stimulation or viewpoint control is introduced, dynamic control is possible, and biomedical materials are moving toward those with two-way rather than one-way functionality. The processing scale of biomaterials decreases to the nano level, the complexity gradually increases, and the functionality becomes complex.

**Figure 2 nanomaterials-12-01377-f002:**
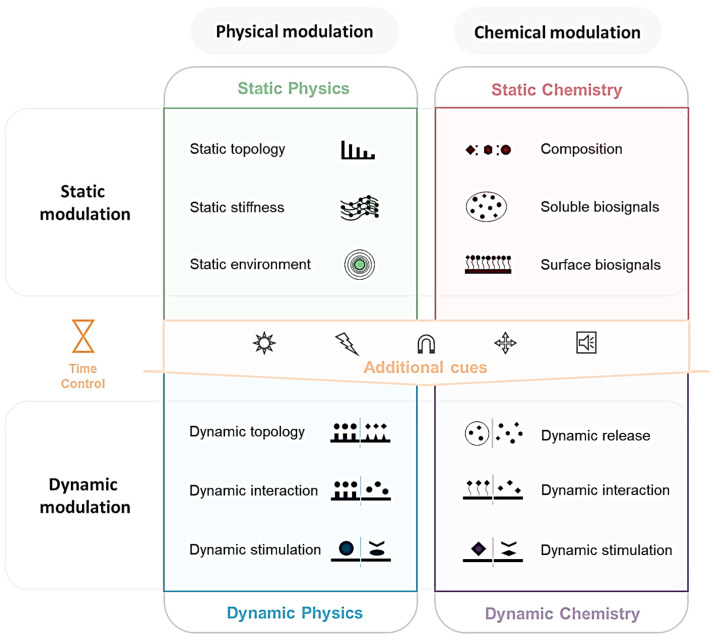
Methods of controlling biomaterials for cell modulation. Biomaterials can be classified as physical-oriented or chemical-oriented depending on their effects on cells. They can then be further classified as static or dynamic. Typical examples of static-physical modulation include static topology, static stability, and static environment (e.g., temperature, electrical/magnetic field), while examples of static-chemical modulation include chemical composition, solid biosignals, and surface-immobilized biosignals. Dynamic modulation enables surface property change, dynamic release, dynamic interaction, and dynamic stimulation through additional cues, such as light, electric/magnetic fields, ultrasonic, and deformation, based on static modulation.

**Figure 3 nanomaterials-12-01377-f003:**
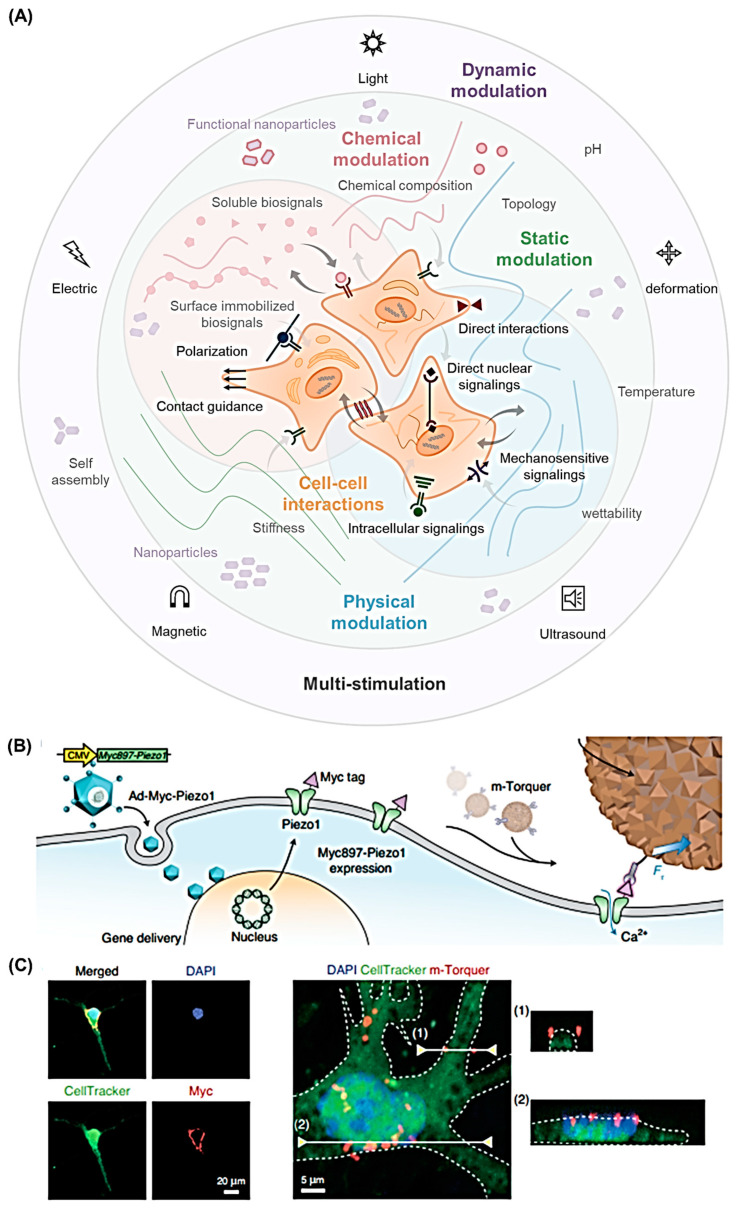
Complex interactions during cellular behavior modulation. (**A**) The physical and chemical properties of the ECM determine cell fates through a variety of mechanisms, such as direct interaction, intracellular signaling, direct nuclear signaling, and mechano-sensitivity signaling. (**B**) Example of dynamic modulation using magnetic stimuli. Schematic of genetic encoding of Piezo1 by Ad-Piezo1 with human cytomegalovirus (CMV) promotor and its magnetomechanical gating with specifically targeted m-Torquer with Myc antibody. Reproduced with permission from [92]. Copyright *Nature Materials*, 2021. (**C**) Confocal microscope images of Piezo1-expressing neuron (DAPI, nucleus; CellTracker, cytosol; Myc, Piezo1, m-Torquer (red); neuron). (1) and (2) are Z-sectioned images. Reproduced with permission from [92]. Copyright Nature Materials, 2021.

**Table 1 nanomaterials-12-01377-t001:** Examples of physical modulation and cell response.

Physical Modulation	Method	Stimulus	Cell Response
Static modulation	Topology	Pit spacing and diameter [50,126]	Cell migration and proliferation [50] Cell proliferation [126]
		Constant ridge and width [51] Spacing interval increased [52] Pitch increase at same depth [129]	Cell migration [51,52] Cellular morphology changes, adhesion, and proliferation [129]
		Nanopillar structures [53] Graded-diameter pillar arrays [54] Square pillars at regular intervals [130] Pyramid pillars with varying nanodiameters [131]	Cell detachment [53] Cell spreading [54] Cellular morphology changes and nuclear deformation [130] Cell adhesion, differentiation, and proliferation [131]
		Concave width and thickness increase [56],	Cell differentiation [56]
		Random convex size [132]	Cell adhesion, migration, and proliferation [132]
	Stiffness	Crosslinker concentration control [64,133]	Cell adhesion, migration, alignment, and protein expression [64] Cellular morphology changes, gene and protein expression [133]
		Aspect ratio change in graded pillar structure [65]	Cell migration [65]
	Environment	Changes in electric field strength [69,107]	Focal adhesion [69] Cellular morphology changes and protein expression [107]
		Homogeneous magnetic field [74,134] Heterogeneous magnetic field [135] Magnetic flux density [136]	Cell alignment [74] Cell adhesion and proliferation [134,135] Cellular morphology changes and differentiation [136]
		Temperature [83,137]	Cell proliferation and differentiation [83] Cellular morphology change [137]
		pH [88,138]	Cell migration, differentiation, and proliferation [88] Cell proliferation [138]
		Wettability, superhydrophobicity [89,90]	Protein adsorption [89] Protein adsorption and cell viability [90]
		Light [139]	Cellular morphology changes and migration [139]
Dynamic modulation	Topology and stiffness	Hydrogel degradation [93,96] Hydrogel crosslinking [94] Photoreversible hydrogel [95] Gradient degradation [97,140]	Gene expression [93] Cell traction and differentiation [94,140] Cell activation [95] Cellular morphology changes and gene expression [96] Cell spreading [97]
		Shape memory polymer [99,141] Temperature-responsive hydrogel [103]	Cell orientation [99] Signal transduction [103] Cell alignment [141]
	Interaction	Hydrophilicity/hydrophobicity control by temperature [101,102,103,142]	Cell sheet engineering [101,102,142] Cell maturation [103]
	Stimulation	Alternating electric field [106,108,109,110]	Cell differentiation and viability [106,109] Cell maturation [108,110]
		Magnetic nanoparticle internalization [111,113,114,115]	Cell differentiation [111] Cell tracking [113] Cell movement manipulation [114,115]
		Pressure and tension [116] Breathing movement [117] Stretching [118,120,126] Compression [121] Loading [122,123] Uniaxial strain [125] Shear stress [126]	Cell orientation [116] Organ on a chip [117] Disease study [118] Tissue transplantation [120] Cell differentiation [121] Tissue formation [122,123] Gene expression [125] Tissue development [126]
		Ultrasound amplitude [127] Piezoelectric effect [128]	Gene expression [127] Cell spreading and focal adhesion [128]

**Table 2 nanomaterials-12-01377-t002:** Examples of chemical modulation.

Chemical Modulation	Modulation Method	Characteristics	Response
Static modulation	Surface chemical properties	Chemical composition changes using SAMs [152,153]	Surface hydrophobicity and protein adsorption [152] Surface hydrophobicity [153]
		Acid treatment using piranha solution [154] Corona discharge treatment [155]	Cell viability, proliferation, and adhesion [154] Surface wettability and cell proliferation [155]
	Soluble biosignals	Growth factor-added media [238]	Proliferation [238]
		Hydrogel carrier [172,239]	Tissue repair and cell proliferation [172] Angiogenesis and tissue repair [239]
		Nanoparticle carrier [173,174] Nanolayered materials [240]	Cell proliferation and wound healing [173,240] Tissue regeneration and Cytocompatibility [174]
		Natural polymer carrier [241,242,243]	Differentiation [241] Proliferation [242] Cell adhesion and growth [243]
	Surface- immobilized biosignals	Physical adsorption [176]	Cell differentiation, adhesion, and proliferation [176]
		Layer-by-layer coating [178,179,244]	Cell differentiation [178,244] Cell morphological changes, proliferation, and adhesion [179]
		Covalent coupling chemistry [185,186,245]	Cell adhesion [185] Cell growth, spreading, migration, and proliferation [186] Cell differentiation [245]
		Covalent immobilization and polymerization chemistry [188]	Cell adhesion [188]
		Ligand spacing [194,195,196]	Differentiation [194,195] Cell adhesion [196]
		Ligand ordering [200,202]	Cell adhesion [200] Proliferation and differentiation [202]
		Ligand recurrence and positioning [201,246]	Cell adhesion and gene expression [201] Differentiation and gene expression [246]
		Ligand aspect ratio [202]	Cell adhesion and gene expression [202]
		Ligand stability [203]	Morphology, differentiation, and adhesion [203]
Dynamic modulation	Dynamic release	Physical revealing of nanoligand and ligand-cation [205,247]	Macrophage regulation [205] Cell differentiation [247]
		Electrical stimulation [207,208] Biomineral degradation [210,211,248,249] Magnetic stimulation [250]	Cell proliferation and differentiation [207,208,210,211,248,249,250]
		Deformation of microporous ferrogels and cell aggregation [209]	In vivo cell and drug delivery [209]
	Dynamic interactions	UV bond cleavage [213,216] Photoisomerization [215,217,251]	Cell adhesion and release [213,215,216,251] Cell adhesion, release, and differentiation [217]
		Polymer cleavage-mediated exposure of photolabile groups [214]	Cell migration, adhesion, and patterning [214]
		Exposure and covering of ligand sites via photoelectrolysis and host–guest interaction [252]	Cell adhesion and detachment [252]
		Electroactive polymer and electric field [218,219]	Cell migration [218] Cell metabolic activity and attachment [219]
		Physical revealing of nanoligand and ligand-cation [19] Macroscale ligand population modulation [221,253] Ligand density control and magnet position control [254,255,256]	Cell spreading [19] Differentiation [19,221] Mechanosensing [19,221,255] Cell adhesion [221,253,254,256] Macrophage regulation [253,254,256]
		In situ metal ion-molecule complexation [220,223]	Cell adhesion and immunoregulation [220] Cell adhesion, mechanosensing, and differentiation [223]
	Dynamic stimulation	Magnetic cell capture [228] Receptor clustering [229,230] Ion channel twisting [257]	Cell migration, differentiation, and targeting [228] Signal transduction [229] Cell targeting and signal transduction [230] Ion channel activation [257]
		Spontaneous linear assembly [232]	Gene delivery and cell targeting [232]
		Tissue stretching [111] Nanoscale stretching-elasticity of ligand sites [222,233,258] Nanoscale vibrations [259]	Cell differentiation [111] Cell polarization [222,258] Cell spreading, and mechanosensing [233] Cell differentiation, adhesion, and mechanosensing [259]
		Electric pulses [224,225]	Cell behavior modulation [224] Cell adhesion [225]
		In situ polymerization on cell surface and enzymatic reactions in cells or tissues [236,237,260]	Cancer therapy [236,237] Neuron regulation [260]
		Ultrasound stimulation for polymer and 3D cyclic mechanical stimulation [234,235,261]	Signal transduction and cancer therapy [234,235,261]

## Data Availability

Not applicable.

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
