# Peer review of "Static and Dynamic Biomaterial Engineering for Cell Modulation"

_nanomaterials, 2022, doi:10.3390/nano12081377_

Round 1

Reviewer 1 Report

A very well-prepared review describing nanoactive biomaterial engineering for cell modulation. Studies on physical modulation (e.g., ECM topography, stiffness, and wettability) as well as chemical manipulation (e.g., composition, and soluble and surface biosignals) have been reviewed in details and carefully summarized.  Review can be published in Nanomaterials mdpi after minor revision.
Following important aspects may be clear.
1. Multi stimuli systems were mentioned in the text very sparingly but these systems probably are most advanced at this time.
2. I think it will be well to provide an appropriate discussion about modulation of the cell differentiation. Authors mentioned often impact nanoactive biomaterials on cell differentiation but the mechanism is not completely clear for Reader. 
3. I suggest to cite some papers where a similar topic was studied. 
https://doi.org/10.1007/s00396-020-04750-0 
https://doi.org/10.1021/acsami.9b18505
And paper (https://doi.org/10.1016/j.apsusc.2017.03.001) where multiple temperature-induced transitions were shown that open perspectives for the application of multiple responsive materials for multiple cellular responses using as a stimulus only temperature.

Author Response

Response to Reviewer 1 comments

A very well-prepared review describing nanoactive biomaterial engineering for cell modulation. Studies on physical modulation (e.g., ECM topography, stiffness, and wettability) as well as chemical manipulation (e.g., composition, and soluble and surface biosignals) have been reviewed in details and carefully summarized.  Review can be published in Nanomaterials mdpi after minor revision. Following important aspects may be clear.

Point 1 : Multi stimuli systems were mentioned in the text very sparingly but these systems probably are most advanced at this time.

Response: We thank the reviewer for raising such major point and agree with the importance of multi stimuli systems for the most up-to-date advancement. Controlling the complex cellular behavior via multi stimuli systems can precisely emulate the intricate native environment as compared to the single stimuli system. To clearly address the importance of multi stimuli systems, we have revised the manuscript especially in the “Conclusions and Perspectives” section.

Point 2 :  I think it will be well to provide an appropriate discussion about modulation of the cell differentiation. Authors mentioned often impact nanoactive biomaterials on cell differentiation but the mechanism is not completely clear for Reader. 

Response: We thank the reviewer for such suggestion that will improve the paper. As the reviewer proposed, clearly defining the mechanism and modulation of cell differentiation via nanoactive biomaterials would kindly allow the readers to follow and understand our paper. Consequently, we have revised the “Conclusions and Perspectives” section of the manuscript to include explanations of cell differentiation by biomaterials such as mechanotransduction and epigenetics with following references added as Ref. 242, 244, and 245.

Point 3 :  I suggest to cite some papers where a similar topic was studied. 

https://doi.org/10.1007/s00396-020-04750-0, https://doi.org/10.1021/acsami.9b18505, and paper (https://doi.org/10.1016/j.apsusc.2017.03.001) where multiple temperature-induced transitions were shown that open perspectives for the application of multiple responsive materials for multiple cellular responses using as a stimulus only temperature.

Response: We thank the reviewer for suggesting excellent references related to the application of multiple responsive materials for multiple cellular responses via temperature. We certainly agree and believe that the technologies reported in these recent papers (Appl. Surf. Sci. 2017, 407, 546-554; Colloid Polym. Sci. 2021, 299, 363–383; ACS Appl. Mater. Interfaces 2020, 12, 5, 5447–5455) will provide deeper understanding and emphasize the importance of multiple responsive materials to the readers. These references have been included in the “Static Topology section” and “Surface Chemical Properties” of revised manuscript as Ref. 52, 141, and 142.

Reviewer 2 Report

It was a manuscript about the application of nanomaterials for the physical and chemical modulation of cells. Here are some comments on this study which should be considered before publication:

  1. There are some grammatical mistakes in the text that should be corrected.
  • "In a study on hole structures conducted by Choi et al. [54], holes with varying diameters were produced by using master"
  • "For example, to study cells in places undergoing constant mechanical motion, such as the lungs, dynamic modulation to apply cyclic mechanical loading to the ECM has been applied in vitro."
  • "small molecules (steroids, phenols, salts), reactive oxygen species (ROS), ions, etc., and vice versa"
  • "Yet, studies on the nature of single surfaces, static modulation, and discontinuous environments should still be carried out for the fundamental understanding of cellular behavior modulation"
  1. Please refer to table 1 and 2 in the main text.
  2. It is better to add a column to the tables for the references.
  3. Please add more figures. You can add the interesting figures of the some of the examples.
  4. Conclusion and perspective part need to be improved
  5. Most of the references are out of date. Please update them.
  6. Maybe it is better to change the title of the manuscript since you didn't just mention the biomaterial

Author Response

Response to Reviewer 2 comments

It was a manuscript about the application of nanomaterials for the physical and chemical modulation of cells. Here are some comments on this study which should be considered before publication:

Point 1 : There are some grammatical mistakes in the text that should be corrected.

- "In a study on hole structures conducted by Choi et al. [54], holes with varying diameters were produced by using master"

- "For example, to study cells in places undergoing constant mechanical motion, such as the lungs, dynamic modulation to apply cyclic mechanical loading to the ECM has been applied in vitro."

- "small molecules (steroids, phenols, salts), reactive oxygen species (ROS), ions, etc., and vice versa"

- "Yet, studies on the nature of single surfaces, static modulation, and discontinuous environments should still be carried out for the fundamental understanding of cellular behavior modulation"

Response: We apologize for any unclarity caused by grammatical errors and complicated sentences. As the reviewer suggested, we have had a native English speaker to thoroughly revise our manuscript for clarity.

Point 2 : Please refer to table 1 and 2 in the main text.

Response: We thank the reviewer for providing such detailed suggestions to improve the conciseness of our paper. In our paper, Table 1 summarized the overall examples related to the physical modulation and following cell responses while Table 2 categorized overall examples related to the chemical modulation and resulting cell responses. As the reviewer proposed, we have referred to the table 1 and 2 in the revised manuscript.

Point 3 : It is better to add a column to the tables for the references.

Response: We apologize for any complexity and confusion caused by addressing the references individually within the table without creating a separate column and have tried making a separate column for the references. However, as various information (e.g., method, stimulus, and cell response) are intertwined with in the tables and references, creating a separate column for references resulted in a more confusing table with less legibility. Instead, we have re-organized and re-classified the tables in our revised manuscript to improve clarity.

Point 4 : Please add more figures. You can add the interesting figures of the some of the examples.

Response: We thank the reviewer for the recommendation that could draw reader’s interest. As the reviewer suggested, we have added interesting figures that serves as representative examples to explain the dynamic modulation mediated cellular regulation as Figure 3B-C. To elaborate, Figure 3B explains the representative mechanism of cellular regulation via magnetic stimuli while Figure 3C shows confocal images of an immune-stained neuron cell after dynamic modulation. We have added figures and revised manuscripts (Figure 3B-C)

Point 5 : Conclusion and perspective part need to be improved

Response: We apologize for the insufficiency of our “Conclusions and Perspectives” section and agree that it should be improved to provide deeper insights. We have significantly revised our manuscript in general with special regards to the “Conclusions and Perspectives” section, by supplementing contents, descriptions, examples, and figures for thorough improvement.

Point 6 : Most of the references are out of date. Please update them.

Response: We apologize for using outdated references and agree that they should be updated for more accurate and up-to-date knowledge. As the reviewer suggested, we have thoroughly updated references in the revised manuscript by deleting outdated papers of low relevance or importance, while adding recent papers (published after 2015) related to multi stimuli responsive materials and control.

Point 7 : Maybe it is better to change the title of the manuscript since you didn't just mention the biomaterial.

Response: We thank the reviewer for highlighting such major point and agree that the title should be changed to precisely represent the paper. We have changed our title as “Static and Dynamic Biomaterial Engineering for Cell Modulation” to accurately address our revised manuscript which discuss about not only the biomaterials, but also the cellular modulations via engineered static and dynamic biomaterials.

Reviewer 3 Report

The work, entitled "Nanoactive Biomaterial Engineering for Cell Modulation," aims to provide an overall perspective on how the biomaterial properties and cellular responses according to the desired biological response in conjunction with complex factors can be actively engineered at the nanoscale to elicit specific cellular responses. It is a deep review and extensive. The paper is well written and has the makings of a publication. A minor point, authors could increase/examples of potential targets.

Author Response

Response to Reviewer 3 comments

Point 1 : The work, entitled "Nanoactive Biomaterial Engineering for Cell Modulation," aims to provide an overall perspective on how the biomaterial properties and cellular responses according to the desired biological response in conjunction with complex factors can be actively engineered at the nanoscale to elicit specific cellular responses. It is a deep review and extensive. The paper is well written and has the makings of a publication. A minor point, authors could increase/examples of potential targets.

Response: We thank the reviewer for suggesting such major advice to improve our manuscript and agree that our paper lacked examples of potential targets at its initial submission. As the reviewer proposed, we have revised the manuscript by adding examples of the potential targets, especially in the “Conclusions and Perspectives” section.

Reviewer 4 Report

This review provides an overall perspective of the current research into biomaterials to inspire the future development of nano-engineered biomaterials for tailoring cells’ bioactivity. The study is interesting and of clear scientific importance for the biomaterials community. However, the review is not new. Fort this reason, the authors should clearly highlight the novelty of their work and the take-home message. The abstract and the conclusions are clear, the “introduction” and the other paragraphs are well structured but more recent references should be added.

Nevertheless, the paper is not suitable for publication in the present form and some minor changes should be properly addressed by the authors to improve the clarity and the understanding.

  • The authors should improve the English; several grammar mistakes make difficult the reading of the paper.
  • Over the past years, different functionalized biomaterials have been extensively studied. However, a large body of literature in this topic is missing. Furthermore, in the “Introduction section”, some of the references are out of date and some more recent ones should replace them.
  • The authors should better underline the novelty of their work. In literature, as already mentioned, there are other papers on this topic. The authors should highlight the main take-home message.
  • The authors should check all the acronyms and cite them the first time they appear in the main text.
  • In my opinion, all the figure captions should be clearer in order to improve the readability and the understanding of the paper.
  • Figures should be improved in terms of quality. Furthermore, figure 2 is not clear. In my opinion, a more complete figure caption could help the reader.

Author Response

Response to Reviewer 4 comments

This review provides an overall perspective of the current research into biomaterials to inspire the future development of nano-engineered biomaterials for tailoring cells’ bioactivity. The study is interesting and of clear scientific importance for the biomaterials community. However, the review is not new. Fort this reason, the authors should clearly highlight the novelty of their work and the take-home message. The abstract and the conclusions are clear, the “introduction” and the other paragraphs are well structured but more recent references should be added. Nevertheless, the paper is not suitable for publication in the present form and some minor changes should be properly addressed by the authors to improve the clarity and the understanding.

Point 1 : The authors should improve the English; several grammar mistakes make difficult the reading of the paper.

Response: We apologize for all the difficulties caused by erroneous use of English such as grammatical errors. For the clarity and better legibility, we have had a native English speaker to thoroughly revise our manuscript.

Point 2 : Over the past years, different functionalized biomaterials have been extensively studied. However, a large body of literature in this topic is missing. Furthermore, in the “Introduction section”, some of the references are out of date and some more recent ones should replace them.

Response: We apologize for the omitted fields and outdated references and acknowledge that these problems could lower the significance and novelty of our paper. Followingly, we have updated the contents and references by deleting outdated papers of low relevance or importance, added up-to-date research (published after 2015), and revised the manuscript to provide more examples and information about the functionalized biomaterials used for application.

Point 3 : The authors should better underline the novelty of their work. In literature, as already mentioned, there are other papers on this topic. The authors should highlight the main take-home message.

Response: We thank the reviewer for pointing out such important problem and agree that the novelty as well as the main take-home message should be highlighted in our paper. The main take-home message of this paper is to emphasize the importance of covering the span from static to dynamic systems where each system is systematically categorized into physical and chemical systems. We also highlight the recently highlighted dynamic systems that regulate complex cellular behavior as a hallmark of biomaterial advancement. As the reviewer suggested, we have changed the title to “Static and Dynamic Biomaterial Engineering for Cell Modulation” and thoroughly revised the manuscript with special regards to “Conclusions and Perspectives” section to underline the novelty and highlight the main take-home message.

Point 4 : The authors should check all the acronyms and cite them the first time they appear in the main text.

Response: We apologize for not explaining the acronyms at their first site. We have thoroughly revised the manuscript to check and explain all the acronyms at their first site.

Point 5 : In my opinion, all the figure captions should be clearer in order to improve the readability and the understanding of the paper.

Response: We thank the reviewer for highlighting this point and agree that the figure captions should be added for its clarity. As the reviewer suggested, we have re-arranged and revised the figure captions to increase legibility and provide clear understanding.

Point 6 : Figures should be improved in terms of quality. Furthermore, figure 2 is not clear. In my opinion, a more complete figure caption could help the reader.

Response: We apologize for the unclarity caused by insufficient figures as well as figure captions and acknowledge that improving those would help the readers understand. As the reviewer suggested, we have revised the manuscript with clearer figure captions, additional figures, and visually improved figures for clarity.

Reviewer 5 Report

The review „Nanoactive Biomaterial Engineering for Cell Modulation” written by Hyung Joon Park et al. is an interesting study and the authors have collected a unique dataset of the current research on biomaterials field. The information presented represents valuable information on how the biophysical, mechanical, and biochemical properties of the extracellular matrix (ECM) influence the cellular response.

Author Response

Response to Reviewer 5 Comments

Point 1 : The review “Nanoactive Biomaterial Engineering for Cell Modulation” written by Hyung Joon Park et al. is an interesting study and the authors have collected a unique dataset of the current research on biomaterials field. The information presented represents valuable information on how the biophysical, mechanical, and biochemical properties of the extracellular matrix (ECM) influence the cellular response.

Response: We appreciate the reviewer for reviewing our paper with such compliments. As the reviewer commented, our paper covered current research on biomaterials field that are engineered with static or dynamic controllability for cellular modulation. We have also explained the reasons and the ways biophysical, mechanical, and biochemical properties of the extracellular matrix (ECM) influence the cellular response. However, our initially submitted paper lacked clarity due to insufficient information, outdated references, and grammatical errors. Followingly, we have had a native English speaker to thoroughly revise our manuscript, and any changes made have been with “Tracking Option”.

Round 2

Reviewer 2 Report

Thanks for addressing the comments.